# Anti-Inflammatory Activity of CIGB-258 against Acute Toxicity of Carboxymethyllysine in Paralyzed Zebrafish via Enhancement of High-Density Lipoproteins Stability and Functionality

**DOI:** 10.3390/ijms231710130

**Published:** 2022-09-04

**Authors:** Kyung-Hyun Cho, Ji-Eun Kim, Hyo-Seon Nam, Dae-Jin Kang, Hye-Jee Na

**Affiliations:** 1Raydel Research Institute, Medical Innovation Complex, Daegu 41061, Korea; 2LipoLab, Yeungnam University, Gyeongsan 38541, Korea

**Keywords:** high-density lipoproteins (HDL), carboxymethyllysine (CML), hyperinflammation, CIGB-258 (Jusvinza), zebrafish

## Abstract

**Background:** Hyperinflammation is frequently associated with the chronic pain of autoimmune disease and the acute death of coronavirus disease (COVID-19) via a severe cytokine cascade. CIGB-258 (Jusvinza^®^), an altered peptide ligand with 3 kDa from heat shock protein 60 (HSP60), inhibits the systemic inflammation and cytokine storm, but the precise mechanism is still unknown. **Objective:** The protective effect of CIGB-258 against inflammatory stress of *N*-ε-carboxymethyllysine (CML) was tested to provide mechanistic insight. **Methods:** CIGB-258 was treated to high-density lipoproteins (HDL) and injected into zebrafish and its embryo to test a putative anti-inflammatory activity under presence of CML. **Results:** Treatment of CML (final 200 μM) caused remarkable glycation of HDL with severe aggregation of HDL particles to produce dysfunctional HDL, which is associated with a decrease in apolipoprotein A-I stability and lowered paraoxonase activity. Degradation of HDL_3_ by ferrous ions was attenuated by a co-treatment with CIGB-258 with a red-shift of the Trp fluorescence in HDL. A microinjection of CML (500 ng) into zebrafish embryos resulted in the highest embryo death rate, only 18% of survivability with developmental defects. However, co-injection of CIGB-258 (final 1 ng) caused the remarkable elevation of survivability around 58%, as well as normal developmental speed. An intraperitoneal injection of CML (final 250 μg) into adult zebrafish resulted acute paralysis, sudden death, and laying down on the bottom of the cage with no swimming ability via neurotoxicity and inflammation. However, a co-injection of CIGB-258 (1 μg) resulted in faster recovery of the swimming ability and higher survivability than CML alone injection. The CML alone group showed 49% survivability, while the CIGB-258 group showed 97% survivability (*p* < 0.001) with a remarkable decrease in hepatic inflammation up to 50%. A comparison of efficacy with CIGB-258, Infliximab (Remsima^®^), and Tocilizumab (Actemra^®^) showed that the CIGB-258 group exhibited faster recovery and swimming ability with higher survivability than those of the Infliximab group. The CIGB-258 group and Tocilizumab group showed the highest survivability, the lowest plasma total cholesterol and triglyceride level, and the infiltration of inflammatory cells, such as neutrophils in hepatic tissue. **Conclusion****:** CIGB-258 ameliorated the acute neurotoxicity, paralysis, hyperinflammation, and death induced by CML, resulting in higher survivability in zebrafish and its embryos by enhancing the HDL structure and functionality.

## 1. Introduction

Hyperinflammation is deeply associated with the cytokine cascade of coronavirus disease (COVID-19) [1], resulting in acute death and chronic pain in autoimmune diseases [2,3], rheumatoid arthritis (RA), psoriasis, and sepsis [4]. The non-enzymatic glycation of blood proteins and carbohydrates produces several advanced glycation end (AGE) products, which have been linked with hyperinflammation to exacerbate the cytokine storm [5]. The glycation of high-density lipoproteins (HDL) is associated with the production of dysfunctional HDL [6], which can aggravate the inflammatory response, even though native HDL exerts antioxidant and anti-inflammatory activity. A comparison of the in vitro antiviral activity showed that native HDL kills SARS-CoV-2, while glycated HDL results in the loss of antiviral activity by decreasing the paraoxonase (PON) activity [7]. Glycated HDL displayed several toxicities in macrophage cells [8], human dermal fibroblasts [9], and zebrafish and its embryos with reactive oxygen species (ROS) production and slower developmental speed [10].

Among the AGE, an elevated serum *N*-ε-carboxymethyllysine (CML) level was also associated with the exacerbation of atherosclerosis via lipoprotein modifications and increased low-density lipoproteins (LDL) susceptibility [11,12]. Patients with type 1 diabetes mellitus (T1DM) showed significantly higher serum levels of CML and high-sensitivity C-reactive protein (CRP) than the control group via an increase in Toll-like receptor 4 (TLR-4) expression in monocytes [13]. The patients also showed remarkably elevated inflammatory cytokines, interleukin (IL)-1β, and tumor necrosis factor (TNF)-α, indicating that an elevated CML level is associated with a pro-inflammatory state [13]. The well-known ligands for TLR-4 are heat shock protein (HSP)-60, endotoxin, such as lipopolysaccharide (LPS) and AGE products including CML [14]. Patients with type 2 diabetes mellitus (T2DM) also showed significantly elevated serum CML and hs-CRP with a remarkable increase in TLR-4 expression and IL-6 and TNF-α, suggesting that high level of CML is associated with pro-inflammatory cytokines [15]. Inhibiting the TLR signaling pathways is an effective therapeutic strategy for suppressing unwanted inflammatory cascades [16]. HDL-like nanoparticles could act as TLR-4 antagonists by sequestering LPS, indicating that HDL inhibits TLR-4 signaling [17]. Reconstituted HDL also exhibited an anti-inflammatory effect via the inhibition of TLR-4 signaling and reduction in TLR-4 expression [18].

CIGB-258 is an altered peptide ligand consisting of 27 amino acids (MW = 2987, isoelectric point = 7.0), derived from HSP60, a 60 kDa chaperone, also known as heat shock protein family D (HSPD) member 1 [19]. HSP60 plays critical roles in the protein synthesis, folding, and delivery of misfolded proteins to proteolytic enzymes in the mitochondrial matrix [20]. This protein has a dual role in immunity because it is known that some HSP60-derived epitopes induced inflammation [21], while others epitopes had anti-inflammatory effects [22]. HSP60 displays several molecular functions in the TLR signaling pathway via apoA-I/HDL binding, ATP binding, protein folding and stabilization, and T-cell activation [23]. CIGB-258, called previously APL-1 or CIGB-814, inhibited the inflammation in several experimental models of rheumatoid arthritis (RA). CIGB-258 reduced the levels of the TNF-α, IL-17 and interferon-γ (IFN-γ) in preclinical studies and in a phase I clinical trial with RA patients [19,24,25]. Likewise, this molecule induced a significant decrease in auto-antibodies against citrullinated self-proteins in RA patients [26]. Furthermore, this peptide increased the frequency of regulatory T cells (T_reg_) and their suppressive capacity against antigen-responding effector CD4+T cells from RA patients [27]. Recently, the intravenous administration of CIGB-258 (1 or 2 mg for every 12 h) showed the therapeutic potential to treat hyperinflammation of patients with COVID-19 with a remarkable decrease in cytokines, which are present in the “cytokine storm”, such as IL-6, TNF-α, and IL-10 [28], but the molecular mechanism of CIGB-258 at the protein level is still unclear.

In the current study, HDL was treated with CML to compare the impairment of HDL quality and functionality in the absence or presence of CIGB-258 and investigate mechanistic insight of the treatment at the molecular basis of protein level. The CML and CIGB-258 were co-injected into adult zebrafish (*Danio rerio*) and its embryos to test amelioration of toxicity and survivability. Zebrafish is a vertebrate model used widely to evaluate the toxicity of AGE and LPS [10]. Zebrafish embryos can be an excellent model to monitor the developmental speed and morphology because zebrafish embryos develop externally. In addition, because zebrafish embryos are optically transparent during development, a microinjection into the embryos is possible, which allows a comparison of the developmental speed and morphology to compare the efficacy of a peptide drug [29]. Regarding the enhancement of HDL stability and anti-inflammatory activity, the efficacy of CIGB-258 was compared with Remsima (Infliximab) and Actemra (Tocilizumab), a TNF-α inhibitor and IL-6 inhibitor, respectively. 

## 2. Results

### 2.1. CIGB-258 Inhibited Glycation of HDL by CML

As shown in Figure 1A, treatment with CML (final 0.2 mM) caused a 1.3-fold higher yellowish fluorescence intensity (FI) during 48 h incubation due to glycation. On the other hand, co-treatment of CIGB-258 (final 20 μM) resulted in a 35% decrease in FI, suggesting that CIGB-258 inhibited the glycation by CML. As shown in Figure 1B, SDS-PAGE showed that the band intensity of apoA-I was decreased significantly by the glycation of CML (lane 2) during 48 h incubation. On the other hand, CIGB-258 treated HDL_3_ showed a 1.8-fold and 1.4-fold more distinct band intensity for 20 μM (lane 3) and 40 μM (lane 4), respectively, even though there was no dose dependency of CIGB-258. 

### 2.2. CIGB-258 Enhanced HDL Stability and Antioxidant Ability

As shown in Figure 2A, TEM image analysis revealed that native HDL_3_ showed a distinct shape and particle morphology (photo a), while CML treated HDL_3_ (photo b) showed a more aggregated and ambiguous particle morphology, indicating that the crosslinking and amyloidogenesis of HDL and apoA-I also occurred by the CML treatment. On the other hand, CIGB-258 co-treated HDL, at both final 20 μM and 40 μM, showed more clear HDL morphology and distinct particle shape than CML-treated HDL_3_ (photos c and d in Figure 2A). CML-treated HDL_3_ showed the smallest particle size, approximately 84 ± 4 nm^2^ with particle aggregation, while native HDL_3_ showed 153 ± 5 nm^2^ size. A co-treatment with CIGB-258 at 20 μM and 40 μM resulted in a larger HDL size, approximately 105 ± 5 nm^2^ (*p <* 0.05) and 122 ± 5 nm^2^ (*p <* 0.01), respectively, indicating that the aggregation of HDL by glycation was reduced in the presence of CIGB-258 (Figure 2B). 

The paraoxonase (PON) assay showed that treatment of CML caused a remarkable loss of PON activity of HDL_3_, around 72% lower PON activity (*p* < 0.01) than with HDL_3_ alone (Figure 2C). On the other hand, co-treatment of CIGB-258 at 20 μM and 40 μM resulted in a 2.5-fold (*p* < 0.01) and 2.7-fold (*p* < 0.01) higher PON activity, respectively, than CML treated HDL_3_. These results indicate that the impairment of HDL_3_ by CML in structural and functional correlations could be improved by the co-presence of CIGB-258. 

### 2.3. CIGB-258 Prevented HDL from Degradation by Iron

As shown in Figure 3A, treatment of ferrous ion (final 120 μM, lane 2) induced severe degradation of HDL_2_ during 72 h at 37 °C compared with HDL_2_ alone (lane 1) as visualized by 0.6% agarose gel. On the other hand, co-treatment of CIGB-258 (lanes 3 and 4) prevented the HDL_3_ from degradation with distinct band intensity in the presence of ferrous ion. Although there was no dose dependency, CIGB-258-treated HDL_3_ showed 2.2-fold and 1.8-fold higher band intensity for 20 and 40 μM of CIGB-258, respectively, than ferrous ion-treated HDL_2_. In addition, ferrous ion-treated HDL showed remarkable aggregated at the loading position, as indicated by the red arrowhead in Figure 3A, but the aggregated band was gradually diminished by a co-treatment with CIGB-258 (lane 3 and 4). 

Measurement of Trp fluorescence showed that ferrous ion-treated HDL_3_ (around 339 nm) showed similar WMF with HDL_2_ alone (around 338 nm), as shown in Figure 3B. On the other hand, co-treatment of CIGB-258 resulted in 6–7 nm more red shift of Trp by 345 nm (final 20 μM treated) and 346 nm (final 40 μM treated) under the presence of ferrous ion, indicating that Trp was more exposed to the aqueous phase by CIGB-258 treatment. These results clearly show that CIGB-258 exerted an anti-glycation effect against CML and a protective effect against proteolysis by ferrous ion. 

### 2.4. Co-Injection of CIGB-258 Protected Embryo from the Toxicity of CML

Microinjection of CML (final 500 ng) into zebrafish embryo caused acute embryo death, with up to 18 ± 4% survival at 24 h post-injection, as shown in Figure 4A, while the non-injected group and PBS injected group showed 89 ± 3% and 72 ± 3% survival, respectively. A co-injection of CIGB-258 (final 1 ng) group showed 58 ± 4% survivability in the presence of CML, suggesting that embryo death by the toxicity of CML was protected by the co-presence of CIGB-258. 

No injection control and PBS-injected control showed a normal developmental speed without defected embryos, as shown in Figure 4B (photos a and b). On the other hand, the CML-injected embryos showed the slowest developmental speed with the highest number of defects, as indicated by the red arrowhead with an abnormal morphology (photo c). By contrast, the CIGB-258 co-injected group showed a higher hatching ratio with faster developmental speed without defect, as shown in Figure 4B (photo d). 

At 26 h post-injection after removing chorion, no injection control and PBS-injected control showed a normal developmental speed and embryo morphology at primordium-6 stage with eye pigmentation and tail elongation with more than 32 somites. On the other hand, however, the CML injected embryo showed a 21-somite stage with the weakest pigmentation in the eye and the shortest tail length elongation, as indicated in the blue arrowhead, while the CIGB-258 co-injected embryo showed around the primordium-6 stage with a stronger extent of pigmentation in the eye and longer tail length than the CML group. The CIGB-258 group showed a similar developmental speed with PBS control and no injection control, indicating that CIGB-258 could neutralize the toxicity of CML in the embryo to recover normal developmental speed and morphology. 

### 2.5. CIGB-258 Ameliorated Acute Paralysis and Death of Zebrafish by CML Injection

An intraperitoneal injection of CML (250 μg in 10 μL PBS) caused acute paralysis in zebrafish until 30 min post-injection. As shown in photo a, Figure 5A, all fish were lying down on the bottom of the cage with occasional trembling without movement or swimming. Although the CML-injected zebrafish could not swim in the bottom of the cage, they were still alive with shuddering until 30 min post-injection, as shown in Appendix A. On the other hand, a co-injection of CIGB-258 (final 1 μg in PBS) and CML caused the faster recovery of swimming ability from paralysis with higher survivability (photo b, Figure 5A). At 30 min of post-injection, 30% of the zebrafish in the CIGB-258 group could swim, as shown in Appendix A, while no fish could swim in CML alone group. The first death of fish appeared at 35 ± 4 min and 50 ± 5 min post-injection in the CML + PBS group and CML+CIGB-258 group, respectively, suggesting that treatment of CIGB-258 attenuated the neurotoxicity and protected the zebrafish from acute death by CML. 

Histology analysis with Hematoxylin and Eosin (H&E) staining showed that CML alone group (photo a) showed a 1.9-fold stronger stained area of the nucleus than the CIGB-258 group (photo b), as shown in Figure 6A. At 60 min post-injection, the CML alone group showed 49% survivability, while the CIGB-258 group showed 97% survivability (*p* < 0.001, Figure 6B). Quantification analysis showed that the CML group showed a 2-fold more infiltration of neutrophils than the CIGB-258 group, indicating that hepatic inflammation was reduced significantly by co-injection of CIGB-258 (Figure 6B). These results suggest that the co-presence of CIGB-258 could induce faster recovery via amelioration of acute paralysis and death by the toxicity of CML in adult zebrafish. 

### 2.6. CIGB-258 and Tocilizumab Protected Acute Death of Zebrafish More Than Infliximab

In order to compare the efficacy of the anti-inflammatory activity with an equal dosage, around final 1 μM, CIGB-258 (1 μg), Infliximab (Remsima, final 43 μg) and Tocilizumab (Actemra, final 44 μg) were injected individually into the zebrafish (around 300 mg of body weight) in the presence of an equal amount of CML (250 μg, final 3 mM in zebrafish body weight). Infliximab and Tocilizumab are TNF-α and interleukin (IL)-6 inhibitors, respectively, and are currently prescribed to treat rheumatoid arthritis. After 30 min post-injection, the Tocilizumab and CIGB-258 groups showed similar numbers and patterns of swimming zebrafish around 23–25%, while the Infliximab group and CML alone group showed around 13% and 0% of swimming zebrafish, respectively, as shown in Figure 7A. After 60 min post-injection, the CIGB-258 group showed the highest number of swimming zebrafish, around 77 ± 6%, while the CML alone, Infliximab, and Tocilizumab groups showed around 20 ± 1%, 37 ± 3% and 67 ± 3%, correspondingly. The CIGB-258 and Tocilizumab groups showed the highest survivability of approximately 97%, while CML alone and Infliximab groups showed lower survivability of approximately 49% and 52%, respectively (Figure 7B). At 30 min post-injection, the zebrafish in the CML alone group were paralyzed and lying down on the bottom of the cage due to the severe neurotoxicity (Figure 7C). On the other hand, the CIGB-258 and Tocilizumab groups showed the faster appearance of swimming fish with a more active pattern of movement. These results showed that the IL-6 inhibitors, CIGB-258 and Tocilizumab, were superior in attenuating the acute inflammation of CML than the TNF-α inhibitor (Infliximab), resulting in faster recovery of swimming ability and higher survivability (Figure 7).

Histology analysis with H&E staining showed that CML alone group (photo a) showed the highest number of infiltrated neutrophils, as shown in red intensity. In contrast, co-injection of the CIGB-258 group (photo d) showed the lowest number of neutrophils, as shown in Figure 8A. The Infliximab group showed a similar H&E-stained pattern to the CML alone group, while the Tocilizumab group showed a similar stained pattern to the CIGB-258 group. The CIGB-258 group and Tocilizumab group showed the smallest H&E-stained area, around 12 ± 1% and 17 ± 2%, respectively, while the CML alone group and Infliximab group showed a more than two-fold larger stained area, approximately 23–24 % (Figure 8B). 

### 2.7. CIGB-258 Group Showed the Lowest Plasma TC and TG Level

After 120 min post-injection, all zebrafish were sacrificed for blood collection in the presence of ethylenediaminetetraacetic acid (EDTA) as an anti-coagulant. The serum lipid profiles showed that the CML alone group had the highest plasma total cholesterol (TC) and TG levels, approximately 182 ± 21 mg/dL and 252 ± 34 mg/dL, respectively, while the CIGB-258 group showed the lowest TC and TG level of approximately 124 ± 21 mg/dL and 145 ± 27 mg/dL, respectively. The Infliximab group showed a similar plasma TC and TG level to the CML alone group of approximately 176 ± 29 mg/dL and 192 ± 38 mg/dL, respectively. The Tocilizumab group showed similar plasma TC and TG levels to the CIGB-258 group, approximately 126 ± 31 mg/dL and 171 ± 40 mg/dL, respectively. These results showed that the CIGB-258 treatment caused a 32% and 43% decrease in serum TC and TG, respectively, compared to the CML alone group. The Tocilizumab group also showed decreased plasma TC and TG: a 31% and 33% reduction, respectively. On the other hand, the Infliximab treatment did not reduce the plasma TC levels, although the plasma TG level was reduced by up to 26%. These results suggest that IL-6 inhibitor, Tocilizumab and the CIGB-258, showed similar pattern of serum lipid profiles, but the TNF-α inhibitor (Infliximab) did not improve the lipid profile and hepatic inflammation, which were caused by the CML treatment in zebrafish.

## 3. Discussion

Glycation stress, such as a fructose treatment, could impair the HDL quality and functionality to cause more atherogenesis, cellular senescence, and embryotoxicity [8,30]. On the other hand, however, there is little information on how CML could impair the HDL quality by accelerating aggregation and the acute inflammatory response and neurotoxicity. To the best of the authors’ knowledge, there are no reports showing that the injection of a small amount of CML (250 μg) causes acute palsy and paralysis of zebrafish resulting in a loss of swimming ability and acute death. This is the first report to show that an intraperitoneal injection of CML could cause acute neurodegeneration with paralysis in adult zebrafish. The neurotoxicity of CML (250 μg) was ameliorated by a co-treatment with CIGB-258 (1 μg) by inhibiting the amyloidogenesis of HDL and recovery of HDL_3_-associated PON activity. HDL exerts anti-atherosclerotic activity with the regression of plaque and potent antioxidant and anti-inflammatory activities [31,32] with a broad spectrum of antiviral and antibacterial activities [33]. HDL has many beneficial qualities and functionalities to prevent infection, inflammation, and incidence of aging-related diseases, such as autoimmune disease, cardiovascular diseases, diabetes, and dementia [34]. 

CIGB-258 protected HDL from the glycation stress caused by CML (Figure 1) and inhibited aggregation by increasing the PON activity (Figure 2). CIGB-258 prevented HDL from proteolytic degradation with the red shift of Trp fluorescence (Figure 3). Embryonic toxicity and death by CML were prevented by a co-treatment with CIGB-258 with the concomitant recovery of normal developmental speed and embryo morphology (Figure 4). The acute neurotoxicity and hepatic inflammation of CML in adult zebrafish were ameliorated by co-presence of CIGB-258 with a faster recovery of the swimming ability and higher survivability (Figure 5 and Figure 6). A co-treatment of CIGB-258 exhibited the highest recovery of swimming ability and survivability (Figure 7) and the lowest hepatic inflammation (Figure 8). The effects were similar to the Tocilizumab group (IL-6 inhibitor), which showed significant improvements over the Infliximab group (TNF-α inhibitor). The CIGB-258 and Tocilizumab groups showed the lowest serum TC and TG levels compared to the Infliximab group and CML alone group (Figure 9). These results suggest that CIGB-258 exhibited a potent anti-inflammatory pharmaceutical effect as an IL-6 inhibitor, rather than the TNF-α inhibitor, via the protective activity of HDL, which exerts antioxidant and anti-inflammatory activity in blood.

It has been reported that glycation was intimately associated with neurodegeneration and neuronal cell death via the aggregation of the Tau protein filament [35]. Immunohistochemical analysis revealed CML in the neurofibrillary tangles (NFT) and astrocytes of patients with progressive supranuclear palsy (PSP), Alzheimer’s disease (AD), and amyotrophic lateral sclerosis (ALS) [36]. Although metabolic syndrome is associated directly with a high risk of cranial nerve palsy from a population-based large-scale cohort study [37], there are no reports showing that an injection of CML could induce acute and short-term cerebral palsy. Although the precise mechanisms are unknown, AGE-mediated neurodegeneration was implicated with inflammation, decreased proteostasis, and glycation of neurotransmitters [38]. As a biomarker of neuroinflammation, a mitochondrial dysfunction stimulated the upregulation of HSP60 and initiated the inflammatory pathways because HSP60 was elevated significantly in diabetic patients [39]. The knockdown of HSP60 decreased mitochondrial activity by increasing cell proliferation [40], and HSP60 could be secreted to the extracellular environment, where it can act as an inflammatory regulator. Furthermore, serum soluble HSP60 was elevated in subjects with atherosclerosis in a general population [41]. Hsp60, a ligand of TLR-4, can bind to TLR-4 on the microglia and activate the TLR-4 signaling pathway. Therefore, HSP60 is an attractive therapeutic target candidate for treating AD because HSP60 protects against Aβ synaptic toxicity by inhibiting toxic oligomerization [42]. In the same context, HDL suppressed neurodegenerative disease because HDL and apoA-I can bind Aβ and inhibit the oligomerization of Aβ and aggregation and protect the neuron system [34]. 

Overexpression of HSP60 restricted the release of mitochondrial dsRNA and ameliorated hepatocellular steatosis and liver inflammation in non-alcoholic fatty liver disease in mice [43]. Interestingly, HSP60 could bind with HDL and apoA-I with high affinity, K_d.HDL_ = 50 nM and K_d.HDL_ = 300 nM, respectively, to exert putative immunoregulatory activity [44]. HDL displays potent antioxidant and anti-atherogenic activity by inhibiting oxLDL uptake into macrophages. On the other hand, HSP60 knockdown enhances oxLDL uptake in macrophages and the induction of inflammatory cytokine, particularly IL-6 [44]. These results suggest that the putative synergistic effect of HSP60 and HDL suppresses the atherogenic and inflammatory process via protein–protein interactions. CIGB-258, an altered peptide ligand from the amino acid sequence of HSP60, also showed the stabilization of HDL particles by inhibiting proteolytic degradation and aggregation, as shown in Figure 1, Figure 2 and Figure 3. Future studies should be necessary to investigate the molecular binding mechanism of CIGB-258 and HDL to induce exposure of intrinsic Trp toward the aqueous phase to maintain protein stability regarding the protective effect in the neuron system. 

Co-injection of CIGB-258 could neutralize the embryotoxicity of CML (Figure 4). This study is the first report to show the developmental toxicity of CML in embryos via direct microinjection, slower developmental speed, more developmental defect, and embryonic death. This result shows a good agreement, at least in part, with the previous report, showing that AGE supplementation in pregnant mice induced neural tube defects by elevating oxidative stress in the mice [45]. On the other hand, there are no reports showing the embryotoxicity of CML by direct injection into embryos in the early developmental stage. An injection of CML into zebrafish embryos caused acute embryo death with remarkable ROS production. This is the first report showing that a co-injection of CIGB-258 induced higher survivability and faster developmental speed than a CML alone injection. 

Interestingly, CIGB-258 and Tocilizumab showed similar efficacy in neutralizing the toxicity of CML (Figure 5 and Figure 6); they share several effects with an IL-6 inhibitor to reduce hepatic infiltration of inflammatory neutrophils and decrease serum TC and TG levels (Figure 7, Figure 8 and Figure 9). Initially, CIGB-258 was developed as a therapeutic to treat RA, namely, APL-1 or CIGB-814, an altered peptide ligand derived from HSP60 [19,24,26]. CIGB-258 is a regulatory CD4+ T cell epitope and induces apoptosis selectively in activated peripheral blood mononuclear cells from RA patients [24]. Results from clinical investigations in RA indicated that this peptide was safe and activated mechanisms associated with induction of tolerance. Probably, therapeutic effect of CIGB-258 in animal models and RA patients is due to the processing and presentation of this altered peptide ligand by dendritic cells to T-lymphocytes could induce the expansion of regulatory T cells. These activated cells migrate to the inflammation site and they could cross-recognize wild-type epitope from HSP60, where it is highly expressed due to the inflammation process. This new contact with HSP60 auto-antigen may induce potent immune-regulatory effect, attenuating auto-reactive T cells responsible of RA pathogenesis and inhibiting inflammatory process [46]. In addition, as evidenced here, this anti-inflammatory effect of CIGB-258 is strongly linked with protective activity on HDL, which exerts antioxidant and anti-inflammatory activity in blood.

As an anti-inflammatory drug, treatment of CIGB-258 caused a decrease in C-reactive protein, ferritin, IL-6, IL-10, and TNF-α in seriously ill and critically ill patients [28]. A treatment of Tocilizumab (Actemra) in adult patients with COVID-19 pneumonia resulted in lower mortality by improving the biochemistry indicators, such as lowered C-reactive protein and ferritin [47]. The mortality of COVID-19 is associated with uncontrolled and systemic inflammation with a cytokine storm that results in catastrophic multiple organ failure. During the hyperinflammation period, common inflammatory mediators (CRP, serum amyloid A, and ferritin), and cytokines (IL-6 and TNF-α) are increased abruptly [48]. Previous reports and the current study show that the CIGB-258 and Tocilizumab might share the mode of action to inhibit the hyperinflammation. In addition, there might be a synergistic effect in CIGB-258 and HDL to inhibit CML toxicity by increasing HDL and apoA-I stability and PON activity, as shown in Figure 1, Figure 2 and Figure 3. 

The over-activation of microglia, in response to certain endogenous toxins, contributes to the progression of several neurodegenerative diseases, including Alzheimer’s disease, Parkinson’s disease, and multiple sclerosis (MS) [49]. Neurodegenerative diseases are associated with the secretion of various pro-inflammatory and cytotoxic factors by activating the microglia in the brain [50]. HSP60 is strongly expressed by activated microglia, and the extracellular release of HSP60 increases the production of other pro-inflammatory factors by binding to TLR-4 and stimulating neuronal cell death [49,50]. HDL and apoA-I suppress inflammation by binding directly to and neutralizing endotoxins, such as lipopolysaccharide, which affect TLR-4 activation and the downstream signaling pathways [51]. Thus, the regulation of activity and expression of HSP60 and HDL might be a potential therapeutic option for treating neurodegenerative disorders [52].

In conclusion, treatment with CIGB-258 neutralized the CML toxicity by enhancing the HDL stability and functionality to prevent embryonic deaths and developmental defects. The treatment of CIGB-258 caused faster recovery from paralysis, which was induced by neurotoxicity of CML, with higher survivability from acute hepatic inflammation. Anti-inflammatory effect of CIGB-258 with protective activity on HDL, represents an attractive therapeutic approach for chronic inflammatory diseases as autoimmune diseases, diabetes, neurodegenerative diseases, and atherosclerosis. 

## 4. Materials and Methods

### 4.1. Materials

*N*-ε-carboxylmethyllysine (CAS-No 941689-36-7, Cat#14580-5g) and ferrous sulfate (CAS-No 7782-63-0, Cat#F7002) were purchased from Sigma–Aldrich (St. Louis, MO, USA). CIGB-258 (Jusvinza^®^), a recombinant peptide from HSP60 with 27 amino acids, in lyophilized powder formula (1.25 mg/vial, Lot# 1125J1/0), was obtained from the Center for Genetic Engineering and Biotechnology (CIGB, Havana, Cuba) under the agreement that it would be used for research purposes only. Remsima (Infliximab 100 mg powder, Celltrion, Incheon, South Korea Lot# 2B3C091) and Actemra (Tocilizumab 400 mg/20 mL, Lot# 21K010E11778, JW Pharmaceutical, Seoul, South Korea) were purchased from Shinsung Pharmaceutical (Seoul, Korea) under the agreement that it would be used for research purposes only. 

### 4.2. Purification of Lipoproteins

LDL (1.019 < d < 1.063), HDL_2_ (1.063 < d < 1.125), and HDL_3_ (1.125 < d < 1.225) were isolated from the sera of young and healthy human males (mean age, 22 ± 2 years), who donated blood voluntarily after fasting overnight by sequential ultracentrifugation. The density was adjusted appropriately by adding NaCl and NaBr, as detailed elsewhere [53], and the procedures were carried out in accordance with the standard protocols [54]. The samples were centrifuged for 24 h at 10 °C at 100,000× *g* using a Himac CP100-NX with a fixed angle rotor P50AT4 (Hitachi, Tokyo, Japan) at the Raydel Research Institute (Daegu, Korea). After centrifugation, each lipoprotein sample was dialyzed extensively against Tris-buffered saline (TBS; 10 mM Tris-HCl, 140 mM NaCl, and 5 mM EDTA (pH 8.0)) for 24 h to remove NaBr.

### 4.3. Glycation of HDL with CML

Glycation was conducted by incubating the purified HDL_3_ (final 2 mg/mL) with 200 μM of CML in 200 mM potassium phosphate/0.02% sodium azide buffer (pH 7.4). The apoA-I content was compared by SDS-PAGE and densitometric analysis because the glycation resulted in a severe decrease in protein content and several beneficial functions of apoA-I and HDL [10,30]. The extent of the advanced glycation reactions was determined by reading the fluorescence intensities at 370 nm (excitation) and 440 nm (emission), as described previously [55]. The BI was compared by band scanning with Chemi-Doc^®^ XR (Bio-Rad) using Quantity One software (version 4.5.2) from three independent SDS-PAGE.

### 4.4. Electron Microscopy

Transmission electron microscopy (TEM, Hitachi, model HT-7800; Ibaraki, Japan) was performed at 80 kV at the Raydel Research Institute (Daegu, Korea). HDL3 was negatively stained with 1% sodium phosphotungstate (PTA; pH 7.4) with a final protein concentration of 0.3 mg/mL in TBS. A 5 μL aliquot of the HDL suspension was blotted with filter paper and was replaced immediately with a 5 μL droplet of 1% PTA. After a few seconds, the stained HDL fraction was blotted onto a Formvar carbon-coated 300 mesh copper grid and air-dried. The shape and size of the HDL were determined by TEM at a magnification of 40,000×, according to previous reports [7].

### 4.5. Paraoxonase Assay

The paraoxonase-1 (PON-1) activity toward paraoxon was determined by evaluating the hydrolysis of paraoxon to p-nitrophenol and diethylphosphate catalyzed by the enzyme [7,56]. Equally diluted CML-treated HDL3 (20 μL, 2 mg/mL), was added to 180 μL of paraoxon-ethyl (Sigma Cat. No. D-9286) containing solution (90 mM Tris-HCl/3.6 mM NaCl/2 mM CaCl2 (pH 8.5)) with or without CIGB-258. The PON-1 activity was then determined by measuring the initial velocity of *p*-nitrophenol production at 37 °C, as determined by measuring the absorbance at 415 nm (microplate reader, Bio-Rad model 680; Bio-Rad, Hercules, CA, USA).

### 4.6. Treatment of Lipoproteins with Ferrous Ion

Ferrous ions (final 120 μM) were administered individually to purified HDL_3_ (1 mg of protein) as previous report [57], followed by incubation for the designated times from 72 h at 37 °C in the presence of 5% CO_2_. After incubation, the lipoproteins were analyzed by electrophoresis (0.6% agarose) to compare the stability of HDL and the apoA-I content. The relative electrophoretic mobility depends on the intact charge and three-dimensional structure of HDL. Hence, agarose gel electrophoresis was conducted with CML-treated HDL_3_ in the presence and absence of CIGB-258 in the non-denatured state, according to a previous report [58].

### 4.7. Wavelength Maximum Fluorescence of HDL

The change in the secondary structure upon treatment with ferrous ion was observed at the wavelengths of maximum fluorescence (WMF) of the tryptophan residues in HDL_3_. The WMF was determined from the uncorrected spectra obtained on an FL6500 spectrofluorometer (Perkin-Elmer, Norwalk, CT, USA) using Spectrum FL software version 1.2.0.583 (Perkin-Elmer) and a 1 cm path-length Suprasil quartz cuvette (Fisher Scientific, Pittsburgh, PA, USA). The samples were excited at 295 nm to avoid tyrosine fluorescence. The emission spectra were scanned from 305 to 400 nm at room temperature.

### 4.8. Zebrafish

Zebrafish and embryos were maintained using the standard protocols [59] according to the *Guide for the Care and Use of Laboratory Animals* [60]. The maintenance of zebrafish and procedures using zebrafish were approved by the Committee of Animal Care and Use of Raydel Research Institute (approval code RRI-20-003, Daegu, Korea). The fish were maintained in a system cage at 28 °C during treatment under a 10:14 h light cycle with the consumption of normal tetrabit (TetrabitGmbh D49304, 47.5% crude protein, 6.5% crude fat, 2.0% crude fiber, 10.5% crude ash, containing vitamin A (29,770 IU/kg), vitamin D3 (1860 IU/kg), vitamin E (200 mg/kg), and vitamin C (137 mg/kg); Melle, Germany).

### 4.9. Microinjection of Zebrafish Embryos

Embryos at one day post-fertilization (dpf) were injected individually by a microinjection using a pneumatic picopump (PV830; World Precision Instruments, Sarasota, FL, USA) equipped with a magnetic manipulator (MM33; Kantec, Bensenville, IL, USA) with a pulled microcapillary pipette-using device (PC-10; Narishigen, Tokyo, Japan). The injections were performed at the same position on the yolk to minimize bias, as described previously [10]. CML (500 ng) alone, CIGB-258 (1 ng), Infliximab (43 ng), and Tocilizumab (44 ng) were injected into flasks of embryos (final volume 5 nL). After injection, the live embryos were observed under a stereomicroscope (Motic SMZ 168; Hong Kong) and photographed (20× magnification) using a Motic cam2300 CCD camera. At 26 h post-injection, each live embryo was compared after removing chorion to compare the developmental stage at higher magnification (50×). 

### 4.10. Injection of CML and CIGB-258 into Adult Zebrafish

Acute inflammation was caused by administering CML (final 250 μg in 10 μL of PBS) by an intraperitoneal injection using a 28-gauge needle into the abdomen region of zebrafish, which were anesthetized by submersion in 2-phenoxyethanol (Sigma P1126; St. Louis, MO, USA) in system water (1:1000 dilution). During 120 min post-injection, the swimming ability and survivability in the CML alone group and CIGB-258 co-injected group were compared because the CML treatment caused acute paralysis, with laying down on the bottom of the cage without swimming ability in the preliminary test. The anti-inflammatory effects of CIGB-258 against the neurotoxicity of CML were compared with Infliximab and Tocilizumab, a TNF-α and IL-6 inhibitor, respectively. After 120 min post-injection, all zebrafish were sacrificed, and blood was collected for analysis.

### 4.11. Plasma Analysis

Blood (2 μL) was drawn from the hearts of the adult fish, combined with 3 μL of phosphate-buffered saline (PBS)-ethylenediaminetetraacetic acid (EDTA, final concentration, 1 mM) and then collected in EDTA-treated tubes. The plasma total cholesterol (TC) and triglyceride (TG) were determined using commercial assay kits (cholesterol, T-CHO, and TGs, Cleantech TS-S; Wako Pure Chemical, Osaka, Japan). Aspartate transaminase (AST) and alanine transaminase (ALT) were measured using a commercially available assay kit (Asan Pharmaceutical, Hwasung, Korea).

### 4.12. Histopathology Analysis

For the morphological tissue observation, some hepatic and muscle tissues were fixed with a 10% formaldehyde solution for 24 h. The samples were then exchanged twice with the same solution, dehydrated with double ethanol, and formatted in paraffin, producing a 5 μm thick tissue slice treated with poly-L-lysine. For morphological analysis, the prepared tissue sections were stained with H&E for the liver tissue at 400 times magnification with an optical microscope. Quantification of nucleus area using Image J software version 1.53r (Bethesda, MD, USA) (http://rsb.info.nih.gov/ij/ accessed on 16 May 2022) to convert the red intensity from H&E staining. 

### 4.13. Statistical Analysis

The data in this study are expressed as the mean ± SD from at least three independent experiments with duplicate samples. For the zebrafish study, multiple groups were compared using a one-way analysis of variance (ANOVA) using a Scheffe test. Statistical analysis was performed using the SPSS software program (version 23.0; SPSS, Inc., Chicago, IL, USA). A *p*-value < 0.05 was considered significant.

## Figures and Tables

**Figure 1 ijms-23-10130-f001:**
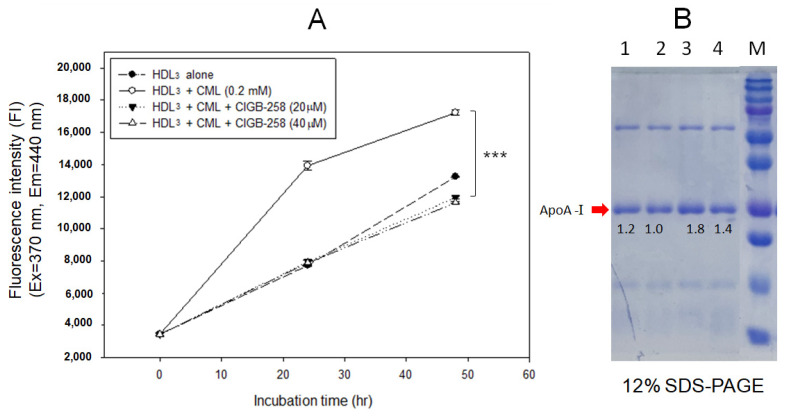
Glycation extent of high-density lipoproteins (HDL) by carboxymethyllysine (CML) in the presence and absence of CIGB-258. (**A**) Measurement of fluorescence intensity (Ex = 370 nm, Em = 440 nm) of HDL_3_ to compare glycated extent during 48 h incubation. *** *p* < 0.001 between HDL_3_ + CML and HDL_3_ + CML + CIGB-258 (20 μM). (**B**) Electrophoretic patterns of the HDL_3_ after 48 h incubation (12% SDS-PAGE). Lane 1, HDL_3_ (1 mg/mL) + PBS; lane 2, HDL_3_ + CML (final 0.2 mM); lane 3, HDL_3_ + CML (final 0.2 mM) + CIGB-258 (final 20 μM); lane 4, HDL_3_ + CML (final 0.2 mM) + CIGB-258 (final 40 μM); lane M, molecular weight marker (Bio-Rad, prestained low-range). The protein bands were visualized by Coomassie Brilliant Blue staining.

**Figure 2 ijms-23-10130-f002:**
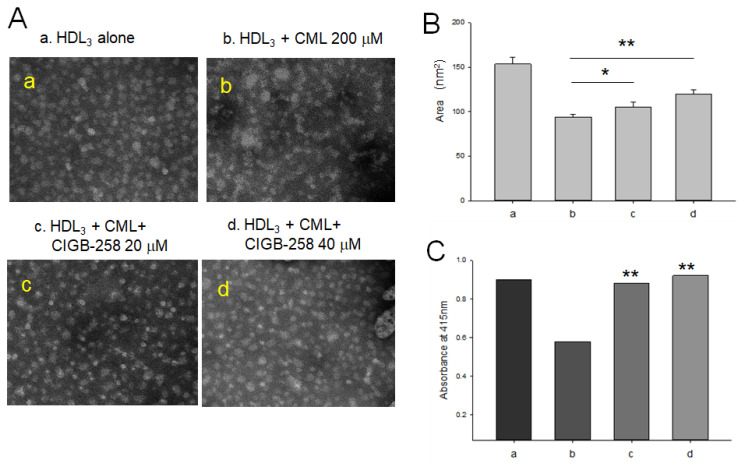
Measurement of high-density lipoproteins (HDL) particle morphology and antioxidant activity in the presence of *N*-ε-carboxymethyllysine (CML) and CIGB-258. a, HDL_3_ (1 mg/mL) alone; b, HDL_3_ + CML (final 0.2 mM); c, HDL_3_ + CML (final 0.2 mM) + CIGB-258 (final 20 μM); d, HDL_3_ + CML (final 0.2 mM) + CIGB-258 (final 40 μM). (**A**). Images of HDL_3_ were visualized from transmission electron microscopy (TEM) after 48 h incubation in the presence and absence of CML and CIGB-258 (40,000×). (**B**). Comparison of HDL_3_ particle size after 48 h incubation under presence and absence of CML and CIGB-258. * *p* < 0.05 between b and c; ** *p* < 0.01 between b and d. (**C**). Comparison of paraoxonase activity from each HDL_3_ after 48 h incubation in presence and absence of CML and CIGB-258. ** *p* < 0.01 versus CML alone (b).

**Figure 3 ijms-23-10130-f003:**
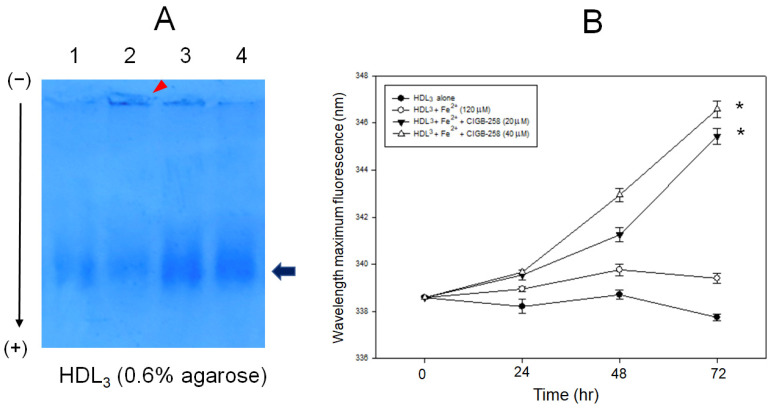
Proteolytic degradation of HDL by addition of ferrous ions (Fe^2+^) and enhancement of HDL_3_ particle stability by the presence of CIGB-258 via the movement of intrinsic Trp fluorescence. (**A**) Electrophoretic patterns of HDL_3_ after 72 h incubation with CML and CIGB-258. Lane 1, HDL_3_ (1 mg/mL) alone; lane 2, HDL_3_ + Fe^2+^ (final 120 μM); lane 3, HDL_3_ + Fe^2+^ + CIGB-258 (final 20 μM); lane 4, HDL_3_ + Fe^2+^ + CIGB-258 (final 40 μM). The red arrowhead indicates the aggregated band by Fe^2+^ at the loading position. The black arrow indicates the HDL particle band intensity after Coomassie Brilliant Blue staining of apoA-I. (**B**) Measurement of wavelength maximum fluorescence of the Trp emission scanning spectrum (Ex = 295 nm, Em = 305–400 nm) during 72 h incubation with HDL_3_ and ferrous ion under the presence of CIGB-258. * *p* < 0.05 versus HDL_3_ + Fe^2+^.

**Figure 4 ijms-23-10130-f004:**
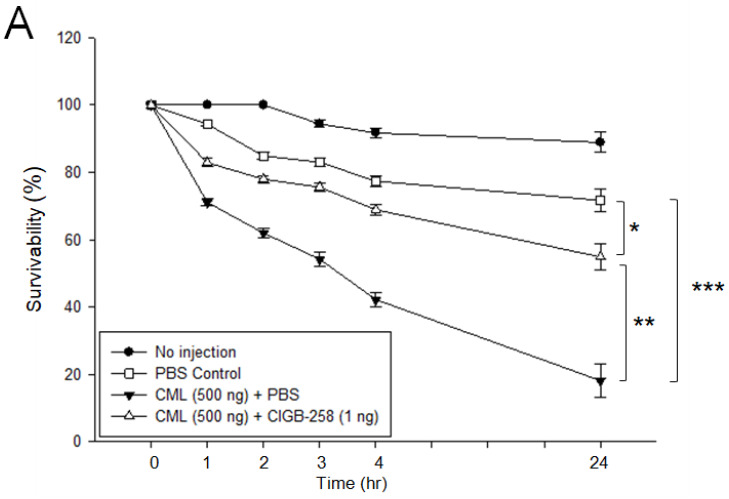
Toxicity of carboxymethyllysine (CML, final 500 ng) in zebrafish embryos in the presence or absence of CIGB-258 (final 1 ng). (**A**) Survivability of the zebrafish embryos during 24 h post-injection. * *p* < 0.05; ** *p* < 0.01; *** *p* < 0.001. (**B**) Stereoimage of the zebrafish embryos at 24 h post-injection. Red arrowheads indicate defected development and death of embryo in CML alone group (photo c). Blue arrowhead indicates the slowest developmental speed in eye pigmentation and tail elongation in CML alone group at 26 h-post injection (photo c).

**Figure 5 ijms-23-10130-f005:**
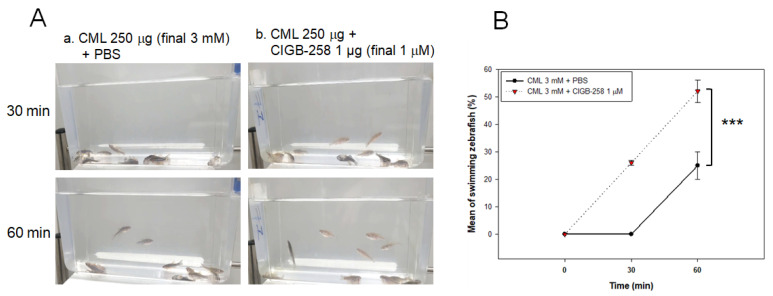
Comparison of the swimming ability after injection of carboxymethyllysine (CML) with or without CIGB-258. (**A**) Still image of swimming pattern of zebrafish after 30 min and 60 min post-injection of CML (250 μg) + PBS (photo a) and CML (250 μg) + CIGB-258 (1 μg, photo b) per fish. (**B**) Percentage of swimming zebrafish after 30 min and 60 min post-injection of CML (250 μg) + PBS and CML (250 μg) + CIGB-258 (1 μg). *** *p* < 0.001.

**Figure 6 ijms-23-10130-f006:**
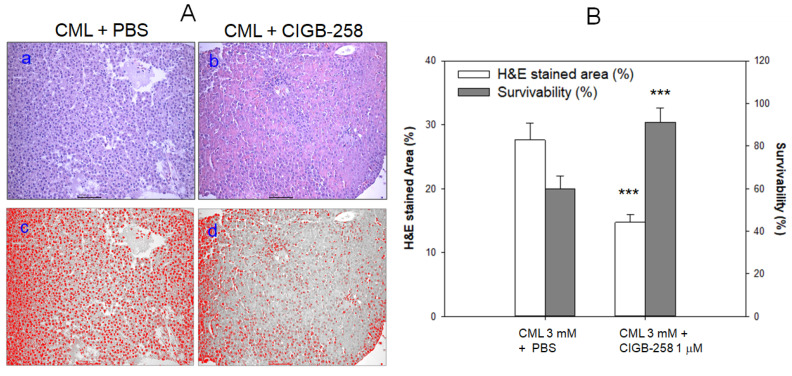
Histologic analysis and survivability of zebrafish injected with carboxymethyllysine (CML) in the presence or absence of CIGB-258. (**A**) Visualization of the infiltration of neutrophils by Hematoxylin and Eosin (H&E) staining as shown in top photographs (photo a and b, 400× magnification). The bottom photographs show conversion of the Hematoxylin-stained area into red intensity (photo c and d, 400× magnification). (**B**) Quantification of the nucleus area from H&E staining using Image J software (http://rsb.info.nih.gov/ij/, accessed on 15 April 2022) as shown in red intensity and survivability of zebrafish at 60 min post-injection. *** *p* < 0.001 versus CML + PBS.

**Figure 7 ijms-23-10130-f007:**
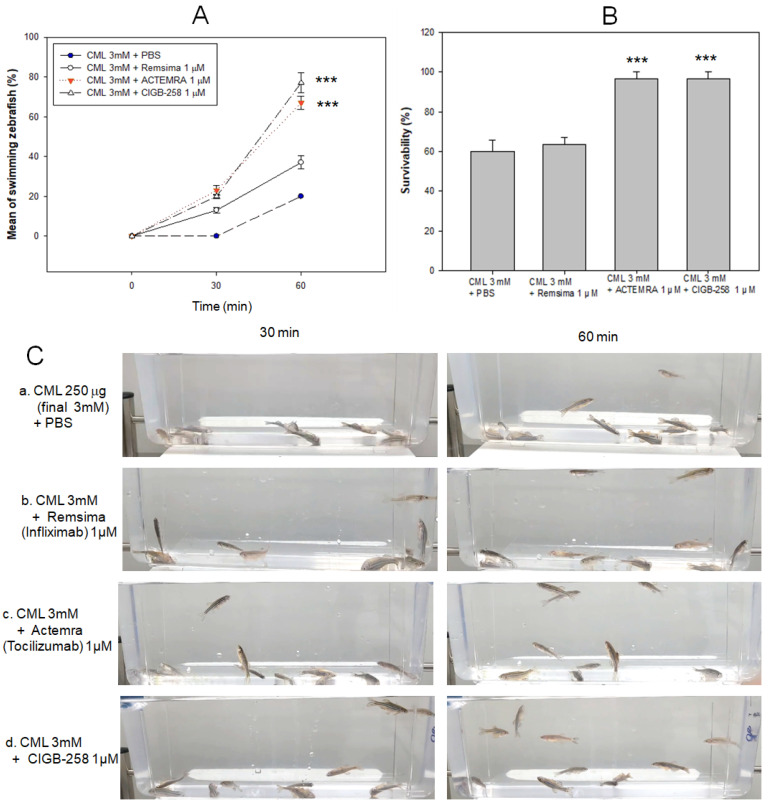
Comparison of the swimming ability and survivability after intraperitoneal injection of carboxymethyllysine (CML) with Infliximab (Remsima), Tocilizumab (Actemra), or CIGB-258 (Jusvinza). (**A**) Percentage of swimming zebrafish after injection of CML with Infliximab (Remsima), Tocilizumab (Actemra), or CIGB-258 at 30 min and 60 min post-injection. *** *p* < 0.001 versus CML + PBS. (**B**) Survivability of zebrafish at 60 min post-injection of CML with Infliximab (Remsima), Tocilizumab (Actemra), or CIGB-258. *** *p* < 0.001 versus CML + PBS. (**C**) Still image of swimming pattern of zebrafish after 30 min and 60 min post-injection of CML with Infliximab (Remsima), Tocilizumab (Actemra), or CIGB-258 at 30 min and 60 min post-injection.

**Figure 8 ijms-23-10130-f008:**
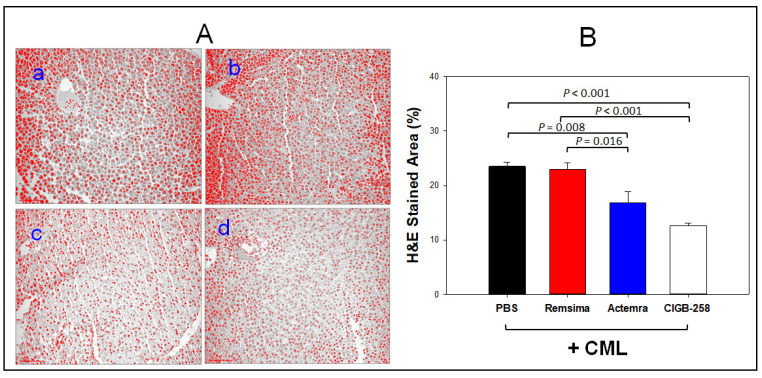
Histologic analysis of hepatic tissue from zebrafish injected with PBS (photo a), Remsima (Infliximab, photo b), Actemra (Tocilizumab, photo c), and CIGB-258 (photo d) in the presence of carboxymethyllysine (CML). (**A**) Visualization of the infiltration of neutrophils by Hematoxylin and Eosin (H&E) staining after converting the Hematoxylin-stained area into a red intensity with 400× magnification. (**B**) Quantification of the nucleus area from the H&E staining using Image J software (http://rsb.info.nih.gov/ij/ accessed on 16 May 2022). Statistical significances of among the groups were indicated as *p* values in the top of graph.

**Figure 9 ijms-23-10130-f009:**
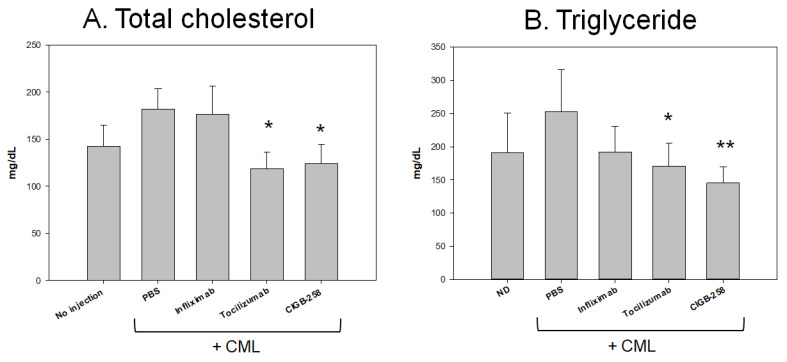
Quantification of lipid profiles from zebrafish injected with PBS, CIGB-258, Infliximab, and Tocilizumab in the presence of carboxymethyllysine (CML). * *p* < 0.05 versus PBS (CML alone) group; ** *p* < 0.01 versus PBS (CML alone) group. All zebrafish were sacrificed at 60 min post injection of each designated drug. (**A**) Quantification of total cholesterol (TC) levels at 60 min post injection. (**B**) Quantification of triglyceride (TG) levels at 60 min post injection.

## Data Availability

The data used to support the findings of this study are available from the corresponding author upon reasonable request.

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
