# Peer review of "Anti-Inflammatory Activity of CIGB-258 against Acute Toxicity of Carboxymethyllysine in Paralyzed Zebrafish via Enhancement of High-Density Lipoproteins Stability and Functionality"

_ijms, 2022, doi:10.3390/ijms231710130_

Round 1

Reviewer 1 Report

In the present study, Cho et al. present a set of results showing the protective effect of CIGB-258 on carboxymethyllysine toxicity through increased high-density lipoprotein stabilization. The study is interesting and relevant, however, the manuscript presents a series of points that detract from its quality.

1. The title is very very long, it complicates the understanding of the message of the results. The reader would have difficulty understanding it.

2. The Summary is extremely long, it abounds in the detail of the results. In addition to not following the recommended structure: Background, objective, methodology, results and conclusion. This section requires substantial improvement.

3. The legends of the figures are not correctly presented, they are incomplete and do not indicate how the data is presented or the type of statistical analysis that is applied.

Author Response

Please find attached doc file for response to reviewer

Reviewer 2 Report

The manuscript describes a large volume of work which appears to have been executed to a high standard. The methods are described comprehensively and the statistical analysis of the results seems rigorous. The standard of presentation is high and the writing style is generally succinct. The following minor points might help the authors revise the manuscript which I have no real hesitation in recommending for publication. 

The abstract is very long and would benefit from some shortening.

Introduction. Are abbreviations like PON and ROS defined somewhere? Beginning of paragraph 3, spelling of isoelectric. 

Author Response

(The authors gave the same response as above.)
